# Evidence for ground state coherence in a two-dimensional Kondo lattice

Wen Wan [1,2], Rishav Harsh[1], Antonella Meninno [3,4], Paul Dreher[1], Sandra Sajan[1], Haojie Guo [1], Ion Errea [1,3,4], Fernando de Juan [1,5] ✉ & Miguel M. Ugeda [1,3,5] ✉

Kondo lattices are ideal testbeds for the exploration of heavy-fermion quantum phases of matter. While our understanding of Kondo lattices has traditionally relied on complex bulk $f$-electron systems, transition metal dichalcogenide heterobilayers have recently emerged as simple, accessible and tunable 2D Kondo lattice platforms where, however, their ground state remains to be established. Here we present evidence of a coherent ground state in the 1T/1H-TaSe$_2$ heterobilayer by means of scanning tunneling microscopy/spectroscopy at 340 mK. Our measurements reveal the existence of two symmetric electronic resonances around the Fermi energy, a hallmark of coherence in the spin lattice. Spectroscopic imaging locates both resonances at the central Ta atom of the charge density wave of the 1T phase, where the localized magnetic moment is held. Furthermore, the evolution of the electronic structure with the magnetic field reveals a non-linear increase of the energy separation between the electronic resonances. Aided by ab initio and auxiliary-fermion mean-field calculations, we demonstrate that this behavior is inconsistent with a fully screened Kondo lattice, and suggests a ground state with magnetic order mediated by conduction electrons. The manifestation of magnetic coherence in TMD-based 2D Kondo lattices enables the exploration of magnetic quantum criticality, Kondo breakdown transitions and unconventional superconductivity in the strict two-dimensional limit.

The complexity of the Kondo lattice problem is best appreciated in comparison with the relative simplicity of that involving a single Kondo impurity. At temperatures below $T_K$, a magnetic impurity coupled to a metal with Kondo exchange coupling $J_K$ starts developing singlet correlations with the conduction electrons and eventually becomes completely screened at T = 0 K. In a periodic array of such impurities, the magnetic moments also develop independent singlet correlations around $T_K$. However, as the temperature is further lowered below a coherence scale $T^*$, the competition between the Kondo exchange and Ruderman–Kittel–Kasuya–Yosida (RKKY) interactions between

moments can drive the system to different ground states like a Kondo paramagnet or magnetically ordered state, as first proposed by Doniach[1] (Fig. 1). A microscopic understanding of the ground state of this coherent Kondo lattice remains a central problem in condensed matter physics, especially as more complex scenarios than those envisioned by Doniach are also possible, where Kondo screening coexists with magnetic order[2]. Understanding the nature and possible types of zero-temperature quantum critical points between such phases and their influence in phenomena like non-Fermi liquid behavior, the Kondo breakdown and fluctuation-mediated

[1]Donostia International Physics Center (DIPC), Paseo Manuel de Lardizábal 4, 20018 San Sebastián, Spain. [2]Materials Genome Institute, Shanghai University, 200444 Shanghai, China. [3]Centro de Física de Materiales (CSIC-UPV/EHU), Paseo Manuel de Lardizábal 5, 20018 San Sebastián, Spain. [4]Departamento de Física Aplicada, Escuela de Ingeniería de Gipuzkoa, University of the Basque Country (UPV/EHU), Plaza Europa 1, 20018 San Sebastián, Spain. [5]Ikerbasque, Basque Foundation for Science, 48013 Bilbao, Spain. ✉e-mail: fernando.dejuan@dipc.org; mmugeda@dipc.org

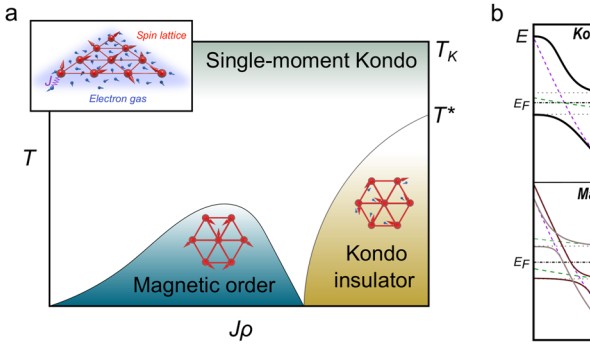
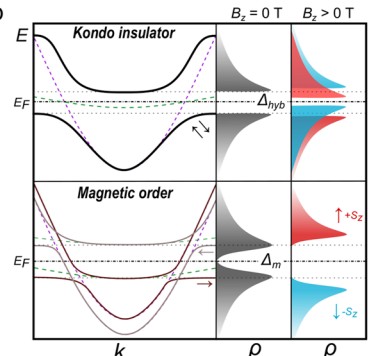

**Fig. 1 | Ground state of a Kondo lattice. a** Doniach phase diagram of a Kondo lattice showing its two possible electronic ground states at low temperatures, the magnetic order and Kondo insulator. The inset shows a representation of the spin lattice (red) and the itinerant electron gas from the metal (blue). $J$ is the local exchange coupling between the localized magnetic moment and the conduction electrons. **b** Schematic band structures and spin-resolved (z component) density of states ($\rho$) of the two ground states. In a Kondo insulator (upper panel), the

conduction band (purple curve) and the localized states (green curve) hybridize to form two spin-degenerate electronic bands (black curves) separated by a gap ($\Delta_{hyb}$). If the Kondo lattice develops magnetic order (lower panel), spin-polarized electronic bands emerge around $E_F$, which leads to two peaks in $\rho$ separated by $\Delta_m$. In the presence of an external magnetic field in the out-of-plane direction, however, $\rho$ develops differently and both ground states can be distinguished.

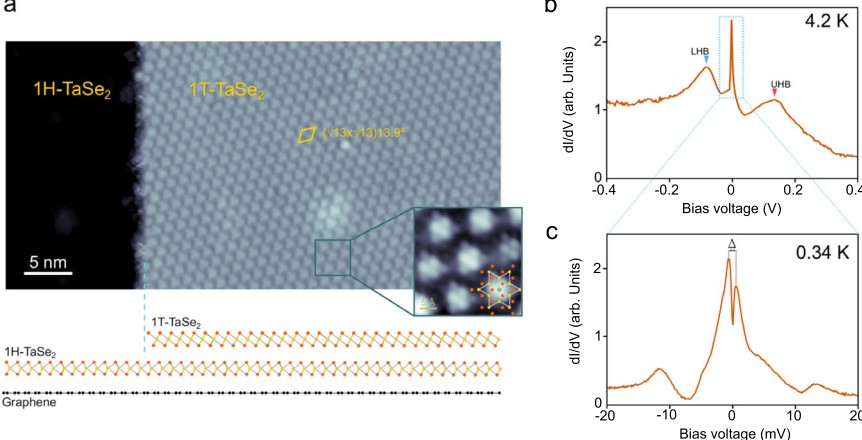

**Fig. 2 | Atomic and electronic structure of the 1T-TaSe₂/1H-TaSe₂ heterobilayer. a** Large-scale STM image of a monolayer of 1T-TaSe₂ on monolayer 1H-TaSe₂ grown on BLG/SiC(0001) ($V_s = -2$ V, I = 0.08 nA, T = 4.2 K). Below a sketch of the vertical arrangement of the atomic layer is shown. Se, Ta, and C atoms are displayed in orange, yellow, and black, respectively. The inset shows a high-resolution STM image of the CDW supercell, where a sketch of the SoD is overlaid ($V_s = -2.5$ mV,

I = 2 nA, T = 0.34 K). **b** Typical dI/dV spectrum taken on the 1T/1H heterostructure at 4.2 K ($V_{a.c.} = 1$ mV). The position of the lower (upper) Hubbard bands are indicated. **c** Low-bias dI/dV spectrum acquired on the 1T/1H heterostructure at our base temperature of 0.34 K showing the emergence of two peaks ($V_{a.c.} = 50\,\mu$V, B = 0 T). The energy separation ($\Delta$) between the peaks maxima is indicated.

superconductivity remains a great challenge in the field. Experimentally, the understanding of these fascinating problems has traditionally been hindered by the complexity and lack of tunability of the *f*-electron compounds like those based in Ytterbium[3] or Cerium[4].

The observation of the Kondo effect in transition metal dichalcogenide (TMD) heterobilayers formed by vertical stacks of T- and H-type monolayers has recently opened a new, simple, and accessible platform to design artificial Kondo lattices[5,6] in strict two-dimensions. In these systems, the 1T monolayer develops a √13 × √13 CDW known as the Star-of-David (SoD) pattern, which leaves a single unpaired electron near the Fermi energy ($E_F$) that forms an effective impurity flat band. The residual Hubbard interaction in this flat band leads to an array of local moments and to a magnetic Mott insulating state. When the 1T layer is stacked onto a metal, the hybridization of the impurity band with the Fermi surface enables the Kondo effect. A narrow Kondo peak has been observed in these T/H heterobilayers of TaSe₂, TaS₂, and NbSe₂ with $T_K$ in the range 18–57 K, providing compelling evidence of the formation of local moments[5–9]. However, lower-temperature evidence for any coherence behavior of the Kondo lattice remains sorely

lacking, and fundamental questions such as the ground state of these compounds in the Doniach diagram remain unanswered.

In this work, we address this problem by performing high-resolution scanning tunneling microscopy/spectroscopy (STM/STS) experiments at 340 mK in the prototype 1T/1H-heterostructure and demonstrate coherent behavior beyond single moment physics, in the form of a split Kondo peak whose separation increases non-linearly upon an out-of-plane magnetic field ($B_z$). Using a periodic Anderson model with parameters extracted from ab initio calculations to compare the signatures of different candidate ground states, we conclude that a magnetic phase of the coherent Kondo lattice with in-plane magnetic order is the most likely scenario consistent with our experiments.

## Results

Our experiments were carried out in high-quality 1T/1H heterobilayers of TaSe₂ grown on epitaxial bilayer graphene (BLG) on 6H-SiC(0001), as sketched in Fig. 2a. The vertical 1T/1H heterobilayer is naturally formed during the growth of few-layer TaSe₂ as both polytypes coexist

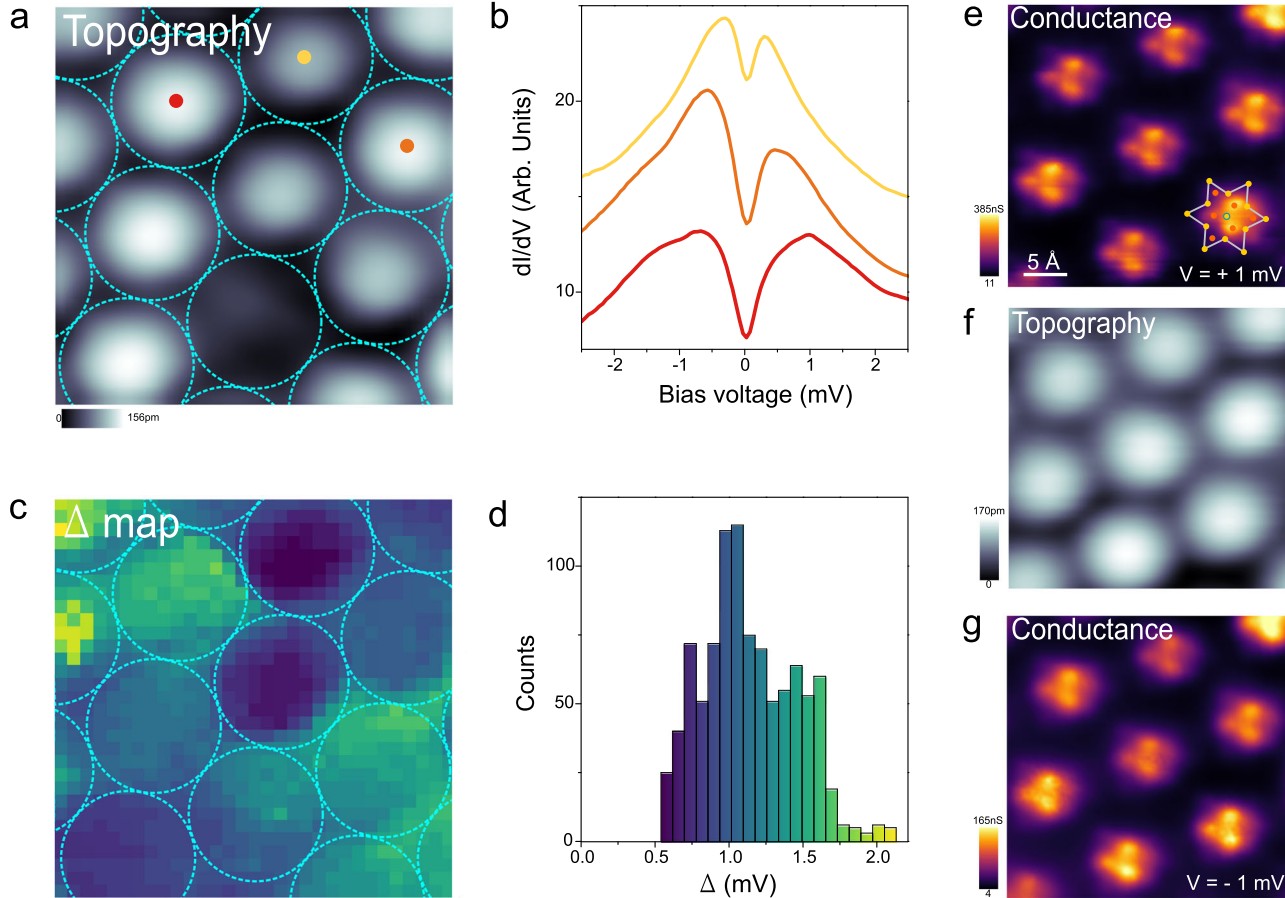

**Fig. 3 | Spatial mapping of the low-energy electronic structure. a** STM topography of the 1T/1H heterostructure ($V_s$ = 4 mV, I = 1.5 nA, T = 0.34 K, 3.3 × 3.3 nm²). The blue circles represent the boundaries of the Star-of-David clusters. **b** dI/dV spectra taken at the center of the three neighboring clusters indicated dots in (**a**) ($V_{a.c.}$ = 40 μV, B = 0 T). **c** Spatially resolved map (32 × 32 pixel, raw data) of Δ (separation of peaks) taken in the region shown in (**a**). The map is extracted from a 40 × 40 dI/dV grid acquired at 0.34 K ($V_{a.c.}$ = 40 μV, B = 0 T). The color scale of the Δ map is shown in the histogram in (**d**). **e** Conductance map (constant height) acquired at $V_s$ = +1 mV and I = 0.16 nA. The atomic position of the SoD cluster is indicated. **f** Topograph of the region where the conductance maps were taken ($V_s$ = −5 mV, I = 0.18 nA). **g** Conductance map at the opposite polarity ($V_s$ = −1 mV, I = 0.2 nA) in the same region (for both conductance maps: $V_{a.c.}$ = 20 μV, B = 0 T, T = 0.34 K).

due to their similar formation energies[10] (see Supplementary Information (SI) for growth details and morphology). The robust commensurate charge density wave (CDW) developed in bulk 1T-TaSe₂ is also easily distinguished in STM images in the monolayer limit as a (√13 × √13)R13.9° triangular superlattice (yellow rhombus in Fig. 2a). Each bright spot in the superlattice corresponds to a cluster of 13 Ta atoms (the SoD) where an individual magnetic moment develops due to the presence of an unpaired electron.

To study the electronic structure of 1T-TaSe₂ on 1H-TaSe₂, we carried out STS measurements at 4.2 K and 0.34 K. dI/dV spectra (dI/dV ∝ LDOS) taken on this heterostructure at 4.2 K (Fig. 2b) are dominated by a prominent single peak centered at the Fermi level ($E_F$) with typical width of 10 meV. The existence of this peak feature, previously reported for single-layer 1T-TaS₂ and 1T-TaSe₂, has been ascribed to a Kondo resonance due to the screening of the SoD magnetic moments by the underlying metal substrate (1H phase)[5,6,8,9]. This peak is flanked by two broader peaks that can be identified as the lower (upper) Hubbard bands, which yield an experimental measure of the Hubbard repulsion of $U$ = 208 ± 4 meV (see SI). As the temperature of the 1T/1H heterostructure is lowered below 4.2 K, the sharp peak feature at $E_F$, however, is better resolved spectroscopically and two symmetric peaks with respect to $E_F$ are observed, as shown in the representative dI/dV spectrum taken at our base temperature of T = 0.34 K in Fig. 2c. It is worth mentioning that the gradual emergence of these two peaks is

not due to a T-dependent process but rather due to an improvement of energy resolution in STS as T is lowered (see SI for the full evolution with T). This double-peak feature is in contrast with the 1T-TaS₂ case, where only one peak appears at sub-kelvin temperatures at zero magnetic field[6].

To gain further knowledge about the low-lying electronic structure of the 1T/1H heterostructure, we carried out spatially resolved measurements at 0.34 K in several nm-sized regions, one of which is shown in Fig. 3a. Figure 3b shows three dI/dV spectra taken at the center of three neighboring clusters (colored dots in Fig. 3a), which reveal that the separation between the two peaks (Δ) presents significant variations. We have quantified these variations by spatially mapping Δ in this region (Fig. 3c) from a 40 × 40 dI/dV grid. As seen, Δ mostly varies between SoD clusters, being largely constant inside each of them. Figure 3d shows the resulting histogram of Δ, which shows a mean value of $\bar{\Delta}$ = 1.2 ± 0.2 mV. Very similar values were found in all the regions studied.

We have also characterized the spatial extension of the two electronic peaks by measuring atomically resolved conductance maps (constant height) of the 1T/1H heterostructure (Fig. 3e–g). Figures 3e, g show two conductance maps taken consecutively at ±1 mV in the same region shown in Fig. 3f. Both conductance maps are nearly identical and reveal that the intensity of these two peaks is mostly localized on the three Se atoms directly bonded to the Ta atom at the center of the

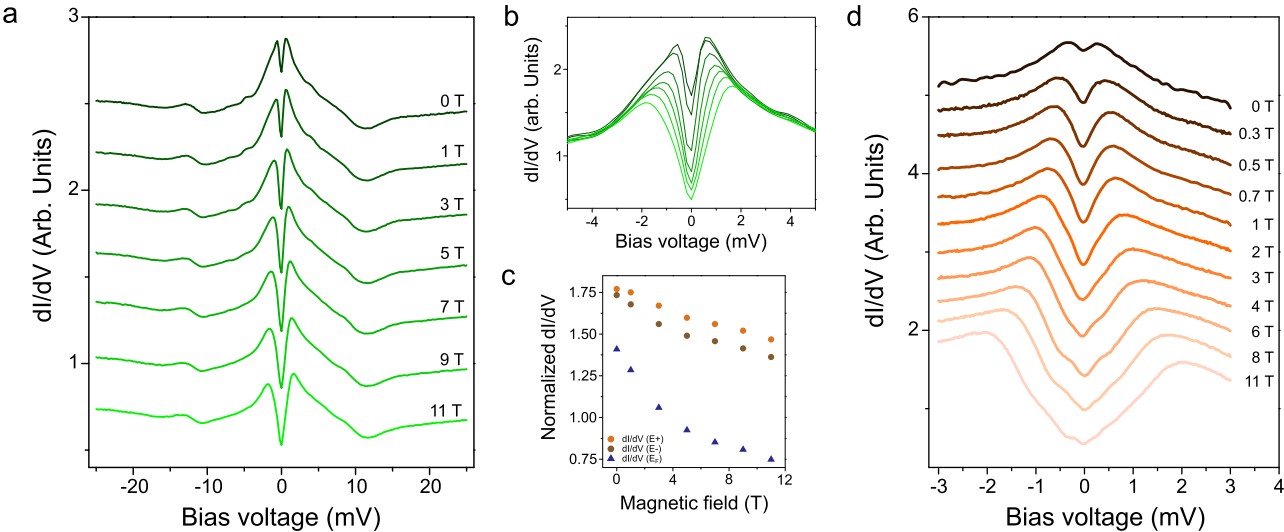

**Fig. 4 | Magnetic-field dependence of the low-energy electronic structure. a** Set of dI/dV spectra taken consecutively at T = 0.34 K as the perpendicular magnetic field is varied in the 0–11 T range ($V_{a.c.}$ = 50 μV). **b** Zoom-in of the set shown in (**a**). The curves are normalized to the set-point voltage ($V_b$ = + 25 mV). **c** Differential conductance values measured at the peaks maxima (E⁺(orange circles) and E⁻ (brown circles)) and $E_F$ (blue triangles) in the dI/dV spectra shown in (**a**). **d** High-resolution series of dI/dV spectra taken consecutively at T = 0.34 K in the vicinity of $E_F$ ($V_{a.c.}$ = 30 μV).

SoD, although some residual intensity lies on the second Se neighbors. This is in good agreement with the expected location of the unpaired electron that gives rise to a net magnetic moment.

In order to elucidate the origin of this two-peak feature, we have performed a thorough characterization of the electronic structure subject to external magnetic fields (out of plane). Figure 4a shows a representative series of dI/dV spectra within ± 25 mV acquired consecutively as $B_z$ is swept up to 11T. The most obvious effect of $B_z$ on the LDOS is the gradual shift of both peaks from $E_F$, which leads to an increase of Δ with $B_z$. In parallel to this energy shift, the LDOS is gradually depleted around $E_F$ and the intensity of the peaks diminish with $B_z$, as shown in the zoom-in of Fig. 4b and plots in Fig. 4c. Beyond these changes, the electronic structure for larger energies seems to be nearly unperturbed by the magnetic field. To confirm that no further features appear in the low-energy electronic structure upon the application of $B_z$, high-resolution dI/dV spectra (-30 μV/point) were systematically taken around the peaks maxima in different regions in the 0–11 T range. As shown in Fig. 4d, no further structures appear within our energy resolution at T = 0.34 K (see SI and ref. 11), and only two peaks remain visible at high magnetic fields.

From our B-dependent STS measurements such as the dI/dV sets of Figs. 4a, d, we have quantified the evolution of the peaks' maxima with $B_z$. Figure 5a shows a typical plot of the peaks' separation Δ as a function of $B_z$. We consistently find that the evolution of Δ with $B_z$ has two differentiated regimes: a strict linear dependence of Δ for $B_z \gtrsim 0.5$ T (linear fit in red, see SI) preceded by a non-linear regime at lower $B_z$ fields (region shaded in green). The linear dependence of Δ with $B_z$ is attributed to a Zeeman splitting with a Landé g-factor of g = 3.1 ± 0.2 (measured from the steepest slope of the dI/dV curves). Figure 5b shows a zoom-in of the low-B field regime, where $Δ(B_z)$ gradually deviates from the linear dependence (beyond the confidence band depicted in gray) below 0.5 T to ultimately reach a deviation of $|S|$ = 0.17 mV at $B_z$ = 0 T from the linear behavior. Another important observation regarding the $Δ(B_z)$ dependence is that this behavior is independent from the polarity of the magnetic field (±$B_z$). This is shown in Fig. 5c (zoom-in in Fig. 5d), where $B_z$ was swept in between ±11 T. As seen, Δ shows an equal behavior with $B_z$ at both polarities, which allow us to rule out hysteresis phenomena and non-symmetric behavior. This symmetric two-regime behavior of $Δ(B_z)$ has been found in most of the SoD clusters where the $B_z$-dependent STS was measured,

from which we can extract an average Δ deviation at zero field of $|S|$ = 0.15 ± 0.02 mV (see histogram in Fig. 5e) and an average field for the non-linear to linear transition at $\bar{B}_T$ = 0.45 ± 0.07 T (see SI for other dI/dV sets).

The observation of a split Kondo peak at zero magnetic field is inconsistent with the behavior of isolated impurities and reveals that the temperature has been lowered enough for lattice coherence effects to set in. Further analysis is nevertheless required to determine the nature of the ground state, and where it lies in the Doniach phase diagram. This analysis can be framed in terms of a periodic Anderson model, where a periodic array of Anderson impurities corresponding to the 1T SoD moments is coupled to the folded metallic bands of the 1H layer

$$H = \sum_{i,j \in H,\sigma} t_{ij}c_{i\sigma}^+ c_{j\sigma} + \sum_{i \in T,\sigma}\left[\epsilon_0 f_{i\sigma}^+ f_{i\sigma} + V(c_{i\sigma}^+ f_{i\sigma} + f_{i\sigma}^+ c_{i\sigma}) + U n_{f\uparrow,i}n_{f\downarrow,i}\right] \quad (1)$$

Here $c_{i\sigma}^+$ creates an electron with spin $\sigma$ in metal site $i$ on the H layer, $f_{i\sigma}^+$ creates an electron on the SoD Wannier orbital $i$ on the T layer with $n_{f\sigma,i} = f_{i\sigma}^+ f_{i\sigma} - \frac{1}{2}$, $t_{ij}$ are the metal hoppings, $\epsilon_0$ is the flat band energy, $V$ is the Kondo hybridization between the $f$ orbital and the $c$ orbital directly below it, and $U$ is the effective Hubbard interaction for the flat band. Since there is one moment per 13 metal atoms, this is a dilute Kondo lattice[12]. In the large $U$ limit where charge fluctuations are frozen, the $f$-level can be represented as a spin ½ exchange-coupled to the metal spins via a Schrieffer-Wolf transformation, giving rise to the Kondo lattice model

$$H = \sum_{i,j \in H,\sigma} t_{ij}c_{i\sigma}^+ c_{j\sigma} + \sum_{i \in T,\sigma} J_K \vec{S}_{f,i} \cdot \vec{S}_{c,i} \quad (2)$$

where $J_K = 2 V^2[1/(U+\epsilon_0)+1/\epsilon_0]$ is the Kondo coupling, $\vec{S}_{f,i}$ is the effective moment at SoD site $i$, and $S_{c,i} = \frac{1}{2}c_i^+ \vec{\sigma}c_i$ is the metal spin directly below the SoD center. The predictions of this model are well known: a Kondo resonance emerges at each impurity at $T_K = W exp(-1/\rho J_K)$, where $\rho$ is the Fermi level DOS and W the metal bandwidth[11,12]. As temperature is further lowered, two types of ground states can be realized: a fully Kondo screened paramagnet is realized at high $J_K$, while different types of magnetic states emerge at low $J_K$, possibly coexisting with partial Kondo screening, with quantum critical points separating these phases

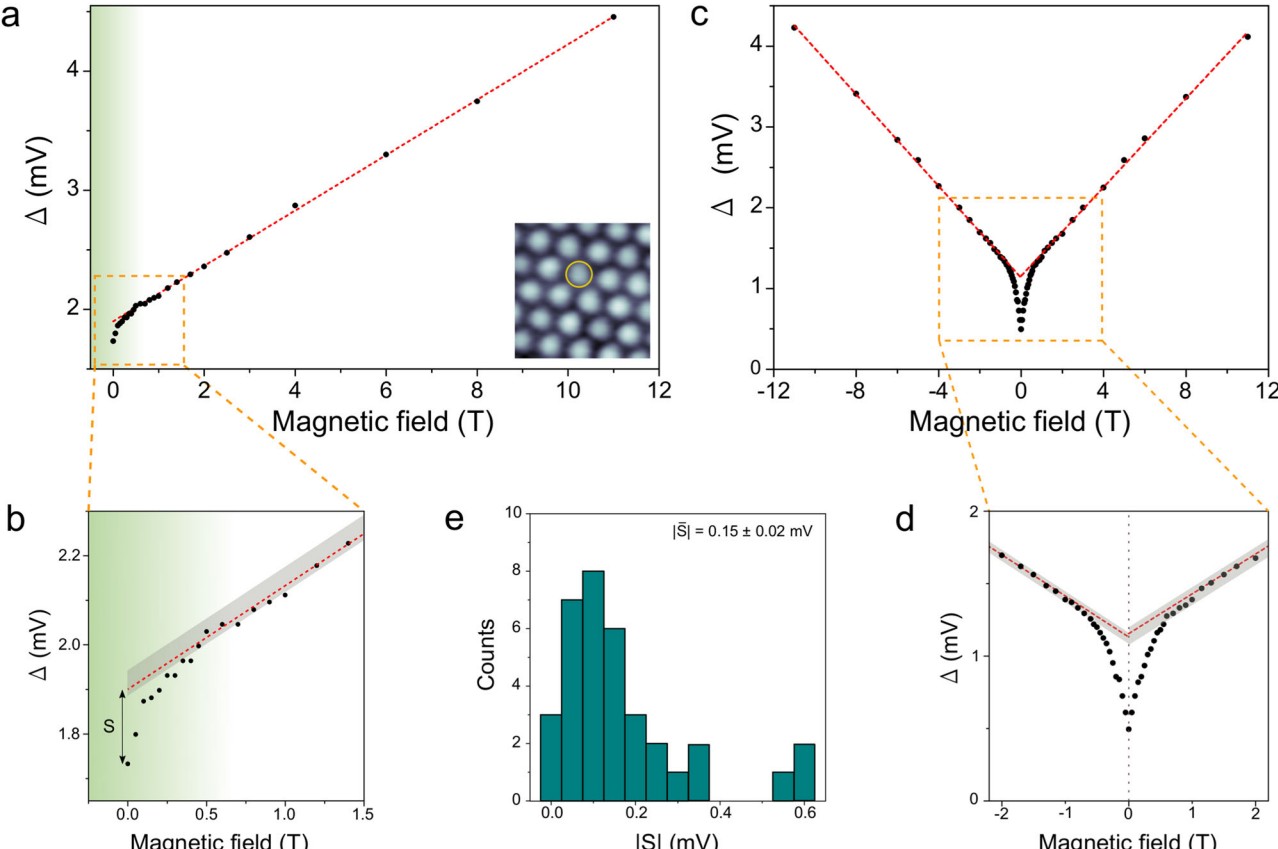

**Fig. 5 | Non-linear behavior of Δ with the magnetic field. a** Plot of Δ as a function of $B_z$ for a series of dI/dV spectra taken consecutively in the SoD cluster shown in the inset. The red dashed line is the linear fit realized in the 1 T < $B_z$ < 11 T. The green shadow indicates the range of $B_Z$ where Δ deviates from a linear relation. The insert shows the SoD (encircled) where the data were taken. **b** Zoom-in of the boxed region in (**a**). The gray region indicates the uncertainty band of Δ in the linear region (see SI). **c** Plot of Δ as a function of $B_z$ for a series of dI/dV spectra taken consecutively in the range ±11 T. The red dashed lines are the linear fits. **d** Zoom-in of the boxed region in (**c**). **e** Histogram showing the occurrence of |S| for the different SoD clusters explored within 0 T < $B_z$ < 11 T.

at values of order $J_K\rho \sim 1$ (refs. 1,13,14). Knowledge of the dimensionless parameter $J_K\rho$ is therefore of key importance to interpret our results.

To estimate the values of the model parameters, we performed ab initio calculations of the band structures for the 1H-, 1T-, and 1H/1T-TaSe₂ structures (see SI). The 1H polytype presents a single half-filled *d*-orbital band at the Fermi level, with a bandwidth of $W = 1.2$ eV and a Fermi level DOS of $\rho = 2.5$ eV⁻¹ (we neglected the known 3 × 3 reconstruction of this band for simplicity, see SI). The 1T polytype displays the √13 × √13 reconstruction with a half-filled flat-band of width 25 meV lying within the CDW gap of 0.55 eV and with its real space spectral weight concentrated in the central atom of the SoD pattern, all consistent with previous reports[15–17]. Our calculation of the 1T/1H heterostructure reveals a set of bands that correspond to those of 1T and 1H sublayers, with a weak hybridization and weak charge transfer from the T to the H layer. This band structure can be fitted with the periodic Anderson model discussed above at $U = 0$ to obtain the metal dispersion and Kondo hybridization $V$. To do so, we first fitted the uncoupled 1H metal band, and then coupled it to a single flat band representing the 1T layer with constant hybridization $V$ which was left as a fitting parameter (see SI for details). Good agreement with the ab initio bands was obtained for $V$ in the range $V = 15$–20 meV. With the value of U obtained experimentally $U = 208 \pm 4$ meV, which is consistent with ab initio estimates for the effective Hubbard $U$ of the flat band in other 1T TMDs[18,19], we finally obtain assuming the symmetric limit of the Anderson model $(\epsilon_0 = U/2), J_K = 8 V^2/U = 8$–15 meV. Taken at face value, the product $J_K\rho = 0.037$ then clearly indicates we should be deep into the magnetic side of the Doniach diagram. An order of magnitude

estimate for $J_K$ can also be obtained from the measured Kondo temperatures[5,6] $T_K = W e^{-\frac{1}{\rho J_K}}$, which yields somewhat larger values $J_{K\cdot}\rho \sim 0.1$, but still in the magnetic side.

Our Kondo lattice model can next be used to rationalize the magnetic-field dependence of the split Kondo peak, which we will argue is also consistent with magnetic order, but not with a Kondo insulator. The essential low-energy features in tunneling to the localized magnetic orbitals are known from mean-field[20–22] and quantum Monte Carlo (QMC) studies of the Kondo lattice[13,14,23,24]. In the mean-field picture, for energies much lower than U, the localized spin is represented by an auxiliary low-energy fermion $\widetilde{f}$ as $\vec{S}_f = \widetilde{f}^+\vec{\sigma}\widetilde{f}$, and the Kondo exchange is decoupled as $J_K\vec{S}_c \cdot \vec{S}_f \rightarrow J_K[\widetilde{V}(\widetilde{f}^+ c + c^+\widetilde{f}) + \vec{S}_f \cdot \langle\vec{S}_c\rangle + \vec{S}_c \cdot \langle\vec{S}_f\rangle]$ with $\widetilde{V} = \langle c^+\widetilde{f}\rangle$ the renormalized Kondo hybridization. This auxiliary fermion band is pinned to $E_F$ and represents the low-energy Kondo peak, which emerges at $T \sim T_K$. Further lowering the temperature leads to a splitting of the Kondo peak of different origins, due to the different finite order parameters $\widetilde{V}, \langle\vec{S}_c\rangle, \langle\vec{S}_f\rangle$. When $J_K > J_C$ this splitting originates from the hybridization $\widetilde{V}$ between the auxiliary fermion band and the metal and leads to a paramagnetic Kondo insulator. If $J_K < J_C$ the splitting rather originates from ferromagnetic (FM) or antiferromagnetic (AFM) order in $\langle\vec{S}_c\rangle$[25,26]. A splitting of the Kondo peak is therefore not sufficient to make a claim about either ground state.

The Zeeman field dependence of the split peak is however very different in each case. In the Kondo insulator, the mean field bands are spin-degenerate, and the Zeeman field shifts the spin up and down bands rigidly in opposite directions[27,28] (see Fig. 1b). This gives rise to four peaks and an eventual closing of the gap[29]. Our resolution clearly

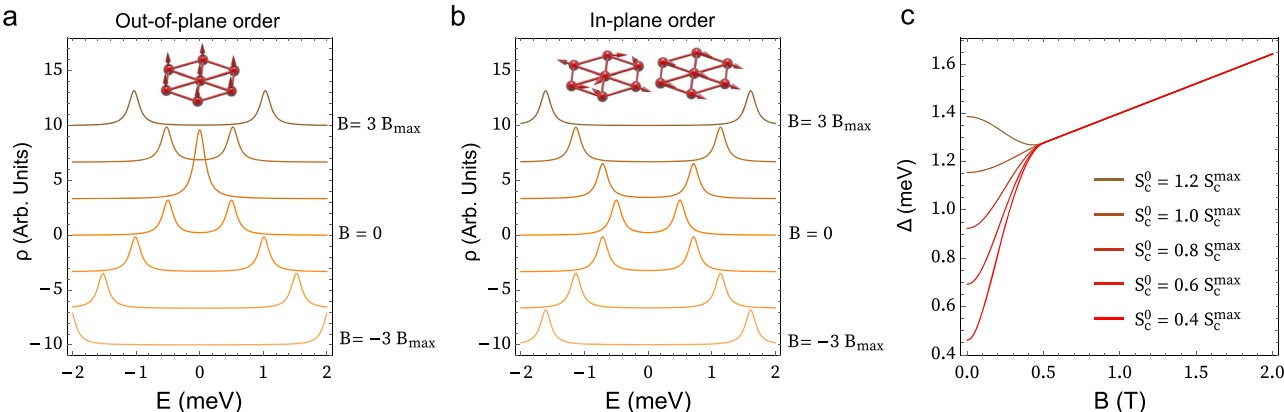

**Fig. 6 | Splitting patterns from mean-field $f$-fermion description. a** Schematic dependence of the spectral function with out-of-plane ferromagnetic order, for different values of the field in integer multiples of $B_{max} = J_K S_c^{max}/g\mu_B$. At low fields, the peak splitting decreases when the field polarity is opposite to the narrow band magnetization, but it increases when they are aligned. **b** Spectral function with in-plane order. The peaks split symmetrically with magnetic field polarity. **c** Non-linear behavior of the gap for the in-plane case due to the reorientation and increased magnitude of the magnetic moment as the field is applied, for $S_c^{max} = 0.077$, $B_c = 0.5$ T and different values of the initial moment $S_c^0$.

allows us to discard this scenario. In the magnetic phase, the main features in the $\widetilde{f}$-fermion spectrum derive from the interplay between the exchange field derived from the metal magnetization $\langle \vec{S}_c \rangle$ and the Zeeman coupling

$$H_{\widetilde{f}} = J_K \frac{1}{2} \widetilde{f}^+ \vec{\sigma} \widetilde{f} \langle \vec{S}_c \rangle - \frac{1}{2}\mu_B g \vec{B} \widetilde{f}^+ \vec{\sigma} \widetilde{f} \qquad (3)$$

where $\mu_B$ is the Bohr magneton and $g = 4.3$ is the measured g-factor. The gap of the $\widetilde{f}$-fermion is obtained as $\Delta = |J_K \langle \vec{S}_c \rangle - \mu_B g \vec{B}|$, from which several general statements can be made, even without knowing the full $B_z$-dependent mean-field solution for $\widetilde{V}, \langle \vec{S}_c \rangle, \langle \vec{S}_f \rangle$. First, the Zeeman coupling polarizes $\langle \vec{S}_f \rangle$, which leads to $\langle \vec{S}_c \rangle$ of opposite sign due to the AFM Kondo coupling. If $\vec{S}_c \parallel \vec{B}$, then the gap depends on $B_z$ as $\Delta = |J_K S_c(B_z) - \mu_B g B_z|$, (with $S_c(B_z) < 0$ for $B_z > 0$). Regardless of the form of $S_c(B_z)$, it is clear that $\Delta$ will not be symmetric in $B_z$ and will be zero when the Zeeman field flips the spin polarization, represented in Fig. 6a for the simplest case of constant $S_c$. However, if $\vec{S}_c \perp \vec{B}$, $\Delta = \sqrt{J_K^2 S_c^2(B_z) + \mu_B^2 g^2 B_z^2}$, and $\Delta$ grows symmetrically with $B_z$ (Fig. 6b, again for constant $S_c$). Our observations are therefore consistent with in-plane magnetic order (FM or AFM since these cannot be distinguished without dispersion for the $\widetilde{f}$-fermion).

The non-linear low-field behavior of the gap can also be understood assuming a smooth interpolating function $\langle \vec{S}_c \rangle (B_z) = (\cos(b) S_c^0, 0, \sin(b) S_c^{max})$ with $b = \frac{\pi B}{2B_c}$, which interpolates between in-plane and out-of-plane order that saturates at $B_c = 0.5$ T. $S_c^0$ can be obtained from the measured $\Delta = 1.2$ meV at zero field and $J_K = 15$ meV as $S_c^0 \sim 0.066$, while $S_c^{max}$ is the intercept of the linear fit with $B = 0$ T (see Fig. 5b), which is on average $\Delta + S = 1.35$ meV and gives the value of $S_c^{max} = 0.077$. With these parameters, we obtain $\Delta(B_z)$ shown in Fig. 6c for different values of $S_c^0$. The qualitative non-linear behavior of the experiment is well reproduced with this model when $S_c^0 < S_c^{max}$, with a faster growth below $B_c$ and a slower one above it. In some instances, we have also observed the opposite behavior (see SI), which is obtained for $S_c^0 > S_c^{max}$ in the model. The non-linear behavior is therefore supportive of a Zeeman-induced transition from in-plane to out-of-plane order.

Within the Kondo lattice model, we have argued that our observed peak splitting indicative of lattice coherence, and more consistent with a magnetically ordered state than with a Kondo

paramagnet. As a further argument that coherence has been reached in our system, we now discard a last possible scenario. If the isolated substrate 1H-TaSe$_2$ was magnetically ordered by itself, isolated moments at SoD sites would display a splitting due to the exchange coupling with a ferromagnet, as observed with isolated spins[30,31]. To put this hypothesis to test, we have measured the Kondo resonance in isolated CoPC impurities placed on top of the 1H-TaSe$_2$ substrate (see SI). A sharp Kondo peak is observed without any splitting at 340 mK, the same temperature at which the 1T/1H structure does show a splitting. This clearly rules out a magnetism in 1H-TaSe$_2$ (consistent with previous X-ray magnetic circular dichroism measurements[5]), and leaves a coherent lattice state as the most likely explanation for the split Kondo peak.

## Discussion

While 1H-TaSe$_2$ is not magnetic, isolated 1H-TMD metals are predicted to be close to a magnetic instability with anomalously large spin susceptibilities, which may contribute to the mechanism selecting in-plane vs out of plane order. For example, isoelectronic NbSe$_2$ shows two leading instabilities, an in-plane AFM (or spin-spiral) with period 4-5 lattice constants[12,32,33], and a subleading fully ferromagnetic state[33,34]. The periodicity of the in-plane AFM state is very close to that of the SoD periodicity, which makes the emergence of in-plane order natural. Nevertheless, the subleading ferromagnetic instability makes the moments relatively soft for tilting out of the plane. Our experiment therefore reveals an interesting connection between the magnetic states of the 1T/1H Kondo lattice and the magnetic fluctuations of the isolated H layer.

Our work demonstrates that the coherence regime in the artificial 2D Kondo lattice realized in 1T/1H TMD heterobilayers can be accessed experimentally. Our observations place the 1T/1H-TaSe$_2$ system likely on the magnetic side of the Doniach phase diagram. The realization of a coherent 2D Kondo lattice with magnetic order enables a model platform where sought-after Kondo phenomenology can be studied with unprecedented resolution and tunability. Several avenues are now opened to fully characterize and tune this versatile Kondo platform, for example by using probes with magnetic sensitivity such as spin susceptibility or spin-resolved STS, which can offer detailed information about the exact magnetic ground state. Furthermore, we expect that the dimensionless parameter $J_K \cdot \rho$ can be tuned by either electrostatic or chemical doping (which can induce large changes in $\rho$ due to the proximity to a van Hove singularity) or by a displacement field, which introduces an interlayer

bias and changes $J_K$ through its dependence on $\epsilon_0$, so that the critical point can be approached. This offers an unprecedented window to access the quantum criticality regime, and potentially to induce unconventional superconductivity[35,36].

## Methods

### Growth of 1T-TaSe₂/1H-TaSe₂ heterobilayers

1T-TaSe$_2$/1H-TaSe$_2$ heterobilayers were grown on BLG/SiC (0001) substrates in a two-step process in our home-made ultra-high vacuum molecular beam epitaxy (UHV-MBE) system under base pressure of ~$2 \times 10^{-10}$ mbar. First, uniform bilayer graphene was prepared by direct annealing 4H-SiC (0001) at a temperature around 1400 °C for 35 min. Second, the as-grown BLG/SiC (0001) substrates were maintained around 550 °C to grow monolayer 1H-TaSe$_2$ by co-evaporation of high-purity Ta (99.95%) and Se (99.999%) with a flux ratio (Ta:Se) of ~1:30. The growth rate for TaSe$_2$ was 2.5 ho/monolayer. Third, the temperature of the substrate was then increased to 640 °C while keeping evaporation parameters unchanged to obtain 1T-TaSe$_2$. After the growth, the samples were kept annealed in Se environment for 2 min, and then immediately cooled down to room temperature. The growth was monitored in situ by reflection high-energy electron diffraction. A ~10-nm-thick Se layer was deposited on the prepared sample before taking it out of UHV conditions for further ex-situ UHV-STM measurements[37,38]. The Se capping layer was subsequently removed in the UHV-STM by annealing at ~300 °C for 40 min. After this process, the typical morphology of our samples, as seen in the STM, is shown in the Supplementary Figure 1.

### STM/STS measurements and tip calibration

STM/STS data were acquired in a commercial UHV-STM system equipped with perpendicular magnetic fields up to 11T (Unisoku, USM-1300). The measurements were carried out at temperatures between 0.34 K and 4.2 K with Pt/Ir tips. To avoid tip artifacts in our STS measurements, the STM tips were calibrated using a Cu(111) surface as reference. We also performed careful inspection of the DOS around $E_F$ to avoid the use of functionalized tips showing strong variations in the DOS. The typical lock-in a.c. modulation parameters for low- and large-bias STS were 20–50 μV and ~1–2 mV at f = 833 Hz, respectively. The energy resolution of the STM instrument at 0.34 K has been tested in bulk Pb(111) before carrying out this experiment. We used Pt/Ir tips for the STM/STS experiments. STM/STS data were analyzed and rendered using WSxM software[39].

### Density functional calculations

We performed ab initio calculations using density-functional theory (DFT) within the plane-wave Quantum Espresso package[40,41] using the Perdew–Burke–Ernzerhof (PBE)[42] parametrization of the exchange-correlation. We have used an ultrasoft pseudopotential with 13 electrons in the valence for Ta and a norm conserving pseudopotential with 6 electrons in the valence for Se. We used a kinetic energy cutoff of 45 Ry and a charge density one of 450 Ry. The structure adopted has a lattice parameter of 3.48 Å in the unit cell of the high symmetry phase and for that phase we used a $40 \times 40 \times 1$ **k** grid and a Methfessel-Paxton smearing of 0.005 Ry for the Brillouin zone integrals[43]. For the super-cell 1T in the CDW phase, we used a denser $24 \times 24 \times 1$ **k** grid to have a good accuracy for the flat band close to the Fermi energy. We assumed the same lattice constant for the H and the T polytypes, and for simplicity, we neglect the $3 \times 3$ CDW of the H phase. The $3 \times 3$ CDW in TaSe$_2$ is weak and does not lead to a dramatic change in the band structure or DOS. Including it in a commensurate cell with the √13 × √13 CDW of the T layer would require a prohibitively large unit cell.

## Data availability

The data that support the findings of this study are available from the corresponding author upon request.

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

## Acknowledgements

M.M.U. acknowledges support by the ERC Starting grant LINKSPM (Grant 758558) and by the grant no. MAT2017-88377-C2-1-R funded by MCIN/AEI/10.13039/501100011033. R.H. acknowledges funding support for project MAGTMD from the European Union's Horizon 2020 research and innovation program under the Marie Sklodowska-Curie grant agreement No 101033538. F.J. acknowledges funding from the grant PGC2018-101988-B-C21 by MCI/AEI/10.13039/501100011033 and by the European Union, and from the Diputación de Gipuzkoa through Gipuzkoa Next (grant 2021-CIEN-000070-01). H.G. acknowledges funding from the EU NextGenerationEU/PRTR-C17.I1, as well as by the IKUR Strategy under the collaboration agreement between Ikerbasque Foundation and DIPC on behalf of the Department of Education of the Basque Government.

## Author contributions

M.M.U. conceived the project. W.W. carried out the MBE growth, the morphology characterization of the samples with the help of R.H., P.D. and S.S. W.W. measured and analyzed the STM/STS data with the help of R.H., P.D., S.S and M.M.U. S.S. and H.G. measured and analyzed the $CoPC/1H\text{-}TaSe_2$ system. A.M. and I.E. carried out the ab initio calculations. M.M.U. supervised the project. F.J. provided the theoretical support and participated in the interpretation of the experimental data. M.M.U. and F.J. wrote the paper with help from the rest of the authors. All authors contributed to the scientific discussion and manuscript revisions.

## Competing interests

The authors declare no competing interests.
