## [Peer Review File · Nature Communications]

REVIEWER COMMENTS

Reviewer #1 (Remarks to the Author):

I thank the authors for submitting a revised manuscript that addresses my comment on the potential tunability of the quantum critical 'knob'.

My other previous comment regarding the presentation of spin-polarised measurements resonates with the main comment of referee two who considers the central claim of this manuscript on the observation of a coherent Kondo lattice speculative. Indeed, this claim is not supported by experimental data, as I wrote before in my report to the Nature Materials submission.

It is great to hear the authors are currently conducting spin-polarised measurements to experimentally establish the presence of long-range magnetic order in this material system. These measurements are required to experimentally verify that the coherent Kondo lattice regime has been reached. These measurements are also required to make the manuscript publishable, if the authors choose to uphold their bold claims ('Our work demonstrates that the coherence regime in the artificial 2D Kondo lattice realized in 1T/1H TMD heterobilayers can be accessed experimentally' and 'The realization of a coherent 2D Kondo lattice with magnetic order...').

I am pleased to recommend publication of this manuscript in Nature Communications, once these data have been presented in a revised manuscript version.

I am broadly happy with the manuscript appearing as is, with two exceptions. The first one is that I do not think the authors have really addressed the question of what experimental evidence for the coherence there is (which is claimed to be there in the title). Without proper experimental evidence, that is something that only appears in the model, and then I think it shouldn't be advertised in the title but rather discussed in the text. I note that reviewer 1 in their first comment allude to the same issue, that the Kondo lattice coherence appears only in the model, but there isn't really experimental evidence that that is what is happening, apart from the spectra looking comparable – but there are likely other scenarios which could result in similar looking spectra (as, e.g., discussed in the previous reports).

While the authors say that 1H-TaSe₂ hasn't been observed to become ferromagnetic in previous studies, it is not clear anyone would have seen that. We are here talking about monolayers, and as the authors state, spin-polarized STM has not been reported so far. NbSe₂ is a very different material, so just because it hasn't been seen to become magnetic has no predictive power for what one might expect in TaSe₂.

Both questions, i.e. re the lattice coherence and/or whether the split features could be a consequence of inelastic excitations, could be addressed by the temperature dependence of the spectra: inelastic features exhibit a specific temperature-dependent broadening, while I guess if the splitting was due to Kondo lattice coherence, the splitting itself should probably have significant temperature dependence? (re the authors reply that one does not expect spin-flip excitations for a spin-1/2 Kondo system, I recommend New Journal of Physics 17, 063016 as reading – it depends on the temperature).

So what I would ask the authors is to either provide a more balanced account for the different scenarios, or provide evidence for the coherence.

The second point is with regards to the resampling in fig. 3c. I am actually a bit stunned that I even need to come back to this point, but below I show a screenshot of the figure shown in 3c on the left and a 32x32 pixel image on the right (the authors claim that they show on image with that resolution on the left). If what was shown on the left was raw data on 32x32 pixels, it should look pixelated as on the right. I will say that this leaves me quite worried about how the authors treat their data, and I can only hope that it is a misunderstanding.

September 6th, 2023

We thank both referees for the constructive and thoughtful reports on our manuscript. While their comments are generally positive, both referees share the main concern that our claim of a coherent magnetic state is not sufficiently supported by our experimental data. The referees suggest additional experiments if our claims were to be maintained or, alternatively, a more balanced account of the different scenarios consistent with our current data. In particular, a plausible scenario is suggested: that we are observing isolated Kondo moments coupled to a magnetic substrate (1H-TaSe₂).

In our resubmitted manuscript, we include a new experiment to put this proposal to test. If monolayer H-TaSe₂ is magnetic, then any Kondo impurity placed on top should show a split Kondo peak due to exchange coupling. Using CoPC molecules as single spins on monolayer H-TaSe₂, we found a sharp Kondo peak that does not split at 340 mK, which rules out a magnetic state for isolated H-TaSe₂. Our result is in good agreement with previous X-ray magnetic circular dichroism (XMCD) measurements carried out in a monolayer of 1H-TaSe₂, where no magnetization was observed (Nature Physics 17, 1154 (2021)).

As explained below, we believe our experiment is more conclusive than spin-polarized STS, which would provide a direct proof of magnetism, but cannot really distinguish between the previous single impurity scenario and a coherent magnetic lattice.

Finally, we do appreciate either way the suggestion of providing a more balanced account of our findings, and the new text includes an enlarged discussion of the different scenarios and our perceived likelihood for them, as well as a modified title and abstract. We hope this revised version will convince the referees of the importance of our work and its suitability for Nature Communications. In the following, we provide a point-by-point response letter to their comments.

Sincerely,

Miguel M. Ugeda and Fernando de Juan, on behalf of the authors.

Response to Reviewer 1

I thank the authors for submitting a revised manuscript that addresses my comment on the potential tunability of the quantum critical 'knob'.

My other previous comment regarding the presentation of spin-polarised measurements resonates with the main comment of referee two who considers the central claim of this manuscript on the observation of a coherent Kondo lattice speculative. Indeed, this claim is not supported by experimental data, as I wrote before in my report to the Nature Materials submission.

Response: Before answering the specific comments of the reviewer, we would like to make the discussion more precise by defining what we mean by lattice coherence (see, for example, Nat. Comm. **9**, 3324 (2018)), which is any signature of physics beyond the behavior of an isolated Kondo impurity. As temperature is lowered, the Kondo peak in the DOS from an isolated impurity just sharpens until its width saturates, and nothing else is observed. In a Kondo lattice, however, when temperature is low enough that impurities interact with each other, the ground state develops lattice coherence and cannot be treated as an array of isolated impurities. Such coherence can be in the form of a Kondo paramagnet (also called a heavy fermion state), or different types of magnetically ordered states. Regardless of the coherent ground state that is reached, the Kondo peak develops fine structure as a result of lattice coherence, typically seen as a splitting of the Kondo peak - a widely accepted fingerprint (see for example Nature **486**, 201 (2012) or Nature **465**, 570 (2010)). The observation of a split Kondo peak in our experiment at zero magnetic field is something that cannot be explained in terms of isolated impurities, and represents the experimental data that supports our claim of ground state coherence.

The only other way to explain that splitting in terms of isolated impurities would be if the substrate (1H-TaSe₂) were somehow intrinsically magnetic, as reviewer 2 suggests. We have conclusively discarded that hypothesis with further measurements (see below). This leaves as essentially the only option that the splitting is due to lattice coherence.

Comment 1: *It is great to hear the authors are currently conducting spin-polarised measurements to experimentally establish the presence of long-range magnetic order in this material system. These measurements are required to experimentally verify that the coherent Kondo lattice regime has been reached. These measurements are also required to make the manuscript publishable, if the authors choose to uphold their bold claims ('Our work demonstrates that the coherence regime in the artificial 2D Kondo lattice realized in 1T/1H TMD heterobilayers can be accessed experimentally' and 'The realization of a coherent 2D Kondo lattice with magnetic order...'). I am pleased to recommend publication of this manuscript in Nature Communications, once these data have been presented in a revised manuscript version.*

Response: Since both referees mention the convenience of performing spin-polarized STM studies to demonstrate magnetism, we believe it is worth discussing this point in detail. SP-STM would indeed be a definitive proof of magnetism, and we agree that currently our claim of a magnetic ground state is inferred from indirect measurements. While as stated in our previous response one of our current long-term directions is to perform SP-STs on this system, these measurements will not be available in any reasonable timescale for the publication of the current work. Because of this, in the revised manuscript we refrain from directly claiming that a magnetic state has been demonstrated.

However, we would also like to argue that SP-STs experiments are not a critical experiment to settle the broader controversy about the ground state of this system. SP-STs can be used to differentiate between the Kondo lattice heavy fermion state (which is not spin polarized) and any magnetically ordered state. However, we have argued that the fact that a split Kondo peak is observed at zero field, and that this gap does not close but rather increases with magnetic field, is clearly inconsistent with a heavy fermion gap (see Refs. 27-29 in the manuscript). SP-STs is therefore not critical to distinguish these two cases. Moreover, SP-STs cannot differentiate between a coherent magnetic state and the scenario provided by reviewer 2 where isolated moments are magnetized by the substrate, since in both cases we expect a split Kondo peak with different spin polarizations on each side. Because of this, we have decided to address this case explicitly measuring single-ion Kondo impurities (isolated CoPC molecules, see more details below), which we believe conclusively proves that H-TaSe₂ is non-magnetic.

In the revised manuscript, we have now revised the reach of our conclusions, especially regarding the demonstration of magnetism in this artificial heterostructure. We now provide a more accurate discussion of the different plausible scenarios, including that suggested by the referee 2. Furthermore, as mentioned before, our new experimental data further supports the conclusion that the peak split at 0.3 K evidences coherence of the spin lattice according to the Doniach formalism.

We hope the sobering of our claims and the additional experimental evidence will convince the referee of the suitability of our manuscript.

Response to Reviewer 2

I am broadly happy with the manuscript appearing as is, with two exceptions.

Comment 1: *The first one is that I do not think the authors have really addressed the question of what experimental evidence for the coherence there is (which is claimed to be there in the title). Without proper experimental evidence, that is something that only appears in the model, and then I think it shouldn't be advertised in the title but rather discussed in the text. I note that reviewer 1 in their first comment allude to the same issue, that the Kondo lattice coherence appears only in the model, but there isn't really experimental evidence that that is what is happening, apart from the spectra looking comparable – but there are likely other scenarios which could result in similar looking spectra (as, e.g., discussed in the previous reports).*

Response: As stated above, we agree with this remark that the evidence for a magnetic state is indirect, and we have been more precise in our claims throughout the manuscript.

Comment 2: *While the authors say that 1H-TaSe₂ hasn't been observed to become ferromagnetic in previous studies, it is not clear anyone would have seen that. We are here talking about monolayers, and as the authors state, spin-polarized STM has not been reported so far. NbSe₂ is a very different material, so just because it hasn't been seen to become magnetic has no predictive power for what one might expect in TaSe₂.*

Response: We agree that up to now, it was not known directly whether monolayer H-TaSe₂ was magnetic or not. We would like to emphasize that previous experiments have reported XMCD on monolayers with a null signal (Nat. Phys. 17, 1154 (2021)), which puts strong constraints on the magnitude of magnetization in this system. But in addition, to address specifically this issue, and the detailed proposal of the referee that what we are observing are isolated Kondo impurities coupled to a magnetic substrate, which would indeed yield this type of zero field splitting as observed on single spins on magnetic clusters (PRB 82, 020406 (2010), PRL 108, 087203 (2012)). To rule out this scenario, we have measured the Kondo fingerprint in the DOS of CoPC molecules, which host isolated spins with Kondo effect for a particular orientation of the molecule, on a monolayer of H-TaSe₂. If H-TaSe₂ is indeed magnetic, the exchange coupling should split the Kondo peak as observed in the mentioned experiments.

The figure below summarizes our main results. The adsorption of individual CoPC molecules on single-layer TaSe₂ leads to two different molecular configurations with respect to the substrate (fig. 1a). These two types of molecules show a markedly distinct low-lying electronic structure. In one case, the molecules show clear signatures of inelastic excitations at ± 25 meV (fig.1b), which have been previously identified as the result of $S = 0$ to $S = 1$ transition (Comm. Phys. 4, 103 (2021), Nano Lett. 19, 4614 (2021), PRL 131, 086701 (2023)).

More relevant for our purpose is the case of the type II molecules, which exhibit a sharp Kondo resonance at E_F , in analogy with the Kondo peak we observe in the T/H phase at 4.2 K (fig. 2b in the manuscript). Unlike in the Kondo resonance in the T/H heterostructure, the Kondo peak measured on the molecule does not split around zero bias as the temperature lowers down to 340 mK. This result was reproduced in tens of individual type-II molecules using a.c. modulations as low as 50 μeV and high density of sampling points (>20 points/mV). The absence of a splitting clearly rules out the existence of magnetism in the H-TaSe₂ substrate. Lastly, the peak splits under magnetic fields as expected for a Kondo resonance.

In summary, these results that we now added to the SI conclusively proves that H-TaSe₂ is not magnetic, and that the splitting of the Kondo peak we observe must originate in the coupled T-H heterostructure. Combined with our previous evidence, and by discarding the rest of scenarios, we believe the most plausible explanation of our experiments is a magnetic state of the type envisioned in the Doniach phase diagram.

Figure 1. **a**, STM topography showing the two configurations of individual CoPC molecules on 1H-TaSe₂. **b** and **c**, typical dI/dV spectra on both types of molecules at $T = 4.2$ K. **d**, dI/dV spectrum on a type-II molecule at $T = 0.34$ K showing no splitting ($V_{\text{a.c.}} = 50 \mu\text{V}$). **e**, Splitting of the Kondo peak due to the Zeeman effect measured at 10 T ($V_{\text{a.c.}} = 50 \mu\text{V}$).

Comment 3: *Both questions, i.e. re the lattice coherence and/or whether the split features could be a consequence of inelastic excitations, could be addressed by the temperature dependence of the spectra: inelastic features exhibit a specific temperature-dependent broadening, while I guess if the splitting was due to Kondo lattice coherence, the splitting itself should probably have significant temperature dependence? (re the authors reply that one does not expect spin-flip excitations for a spin-1/2 Kondo system, I recommend New Journal of Physics 17, 063016 as reading – it depends on the temperature). So what I would ask the authors is to either provide a more balanced account for the different scenarios, or provide evidence for the coherence.*

Response: To discuss this issue, we will briefly review our comments about inelastic spin-flip excitations to ensure our explanation is conveyed effectively. In our previous report, when we mentioned that inelastic spin-flip excitations can only occur for $S > 1/2$, we were referring to the case at zero magnetic field, where magnetic anisotropy terms give rise to different energy levels of the magnetic atom, and when the bias is larger than the transition energy a new channel for tunneling is available which gives to a step in the conductance. For example, this is observed for $S = 3/2$ with Co as mentioned, where both a Kondo peak and a higher energy step are observed. At zero magnetic field and for $S = 1/2$, there is simply no magnetic anisotropy and no extra steps, and only a Kondo peak is observed.

In the presence of magnetic field for $S = 1/2$, as explained in the recommended reading (NJP 17, 063016 (2015)), two features emerge: first the Kondo peak splits, and second a new inelastic step is found at the same position of the split peak. The addition of these two features results in a peak feature which has a steep rise and a much slower decline, and for large fields the contribution from the step dominates. We do agree with the reviewer that inelastic scattering contributes to the conductance, but what we meant to explain is that inelastic excitations do not represent an independent, alternative explanation of the data. Rather, the splitting of the Kondo peak and the opening of an inelastic channel occur simultaneously and effectively give rise to a splitting of the Kondo peak with a more complex lineshape as discussed.

Importantly, the opening of the inelastic channel would occur for a single impurity coupled to a ferromagnetic substrate, as well as for a coherent magnetic state in the limit where the splitting of the pseudo-fermion band level is larger than its dispersion. The Kondo phenomenology would be very similar: the Kondo peak is split at zero field, and the application of a magnetic field further splits the peak, which slowly evolves towards a step function at large fields. The main difference is the origin of the magnetism: for the isolated impurities the substrate is already magnetic, but for the magnetic transition in the Doniach phase diagram the partially screened moments couple to each other and establish long range magnetic order. Regarding the temperature dependence, we acknowledge that both the Kondo signal and the inelastic signals should be temperature dependent, a guess we share with the

reviewer, but we have not been able to conceive a hypothesis that would distinguish between the two scenarios.

To summarize, we believe the alternative scenario of a single impurity coupled to a ferromagnetic substrate should be taken into consideration, but not because of its contribution to inelastic scattering, which is present in both cases and is required to understand the detailed peak shape but not its overall position.

Incidentally, the detailed shape does matter for example when estimating the g factor from the field dependent splitting. Because of the opening of an additional inelastic scattering channel, the maximum of the peak is not a good indicator of the Zeeman energy, which is actually closer to the steepest slope point in the peak. We have redone the analysis of the peak splitting taking this into account and found a reduced $g \sim 3$, which is now reflected in the manuscript.

We thank the referee for the expert comments on inelastic scattering and the alternative scenario with a magnetic substrate, which helped to significantly improving our manuscript.

Comment 4: *The second point is with regards to the resampling in fig. 3c. I am actually a bit stunned that I even need to come back to this point, but below I show a screenshot of the figure shown in 3c on the left and a 32x32 pixel image on the right (the authors claim that they show on image with that resolution on the left). If what was shown on the left was raw data on 32x32 pixels, it should look pixelated as on the right. I will say that this leaves me quite worried about how the authors treat their data, and I can only hope that it is a misunderstanding.*

Response: The referee is right here. We contacted the developers of the WSxM software used to analyze and render the STM/STS data, who confirmed us that 2D images are shown linearly interpolated by default. However, this option can be disabled as shown in the image below. We have now replaced the interpolated map of figure 3c by the raw one in the revised manuscript.

REVIEWERS' COMMENTS

Reviewer #1 (Remarks to the Author):

I was pleased to read the revised manuscript and I appreciate the provision of additional experimental data to rule out alternative scenarios. All of my comments were addressed and I can recommend publication of this manuscript without further revisions in Nature Communications.

Reviewer #2 (Remarks to the Author):

I appreciate that the authors have made an effort to address the question of coherence vs coupling of effectively independent Kondo impurities to a ferromagnetic layer, e.g. if the 1H-TaSe₂ layer did become ferromagnetic by providing additional measurements. While the CoPc-molecules do suggest that the 1H-TaSe₂ by itself is not magnetic, the sensitivity of correlated systems to minor structural changes means that a 1H-TaSe₂ underneath a CoPc molecule might well behave differently than underneath 1T-TaSe₂. So am I convinced by the evidence? Not fully, but then I am happy to leave that question to be resolved by the scientific record and follow-up work.

It is good to finally see the data without interpolation shown in fig. 3c.